# The prevalence of poor sleep quality and associated risk factors among Chinese elderly adults in nursing homes: A cross-sectional study

Xidi Zhu[1☯], Zhao Hu[1☯], Yu Nie[1], Tingting Zhu[2], Atipatsa Chiwanda Kaminga[3,4], Yunhan Yu[1], Huilan Xu[1]*

1 Department of Social Medicine and Health Management, Xiangya School of Public Health, Central South University, Changsha, China, 2 Department of Scientific Research Management, Shanghai Health Development Research Center, Shanghai, China, 3 Department of Mathematics and Statistics, Mzuzu University, Mzuzu, Malawi, 4 Department of Epidemiology and Health Statistics, Xiangya School of Public Health, Central South University, Changsha, Hunan, China

☯ These authors contributed equally to this work.
* xhl6363@sina.com

**Data Availability Statement:** Data for this study contain potentially identifying or sensitive patient information. Therefore, it would be available upon reasonable request from the Ethics Committee of

## Abstract

### Background

Sleep problems have become the most common complaints among the elderly. There are a few studies that explored the prevalence of poor sleep quality and its associated factors among the elderly in nursing homes. Therefore, this study aimed to examine the prevalence of poor sleep quality and its associated factors among the Chinese elderly in nursing homes.

### Methods

A total of 817 elderly residents, from 24 nursing homes, were included in this cross-sectional study. Sleep quality was assessed using the Pittsburgh Sleep Quality Index (PSQI), and poor sleep quality was defined as PSQI >5. Multiple binary logistic regression was used to estimate the strength of the association between risk factors and poor sleep quality in terms of adjusted odds ratios (AORs) and their 95% confidence intervals (CIs), and interactions of risk factors for poor sleep quality were also examined.

### Results

The prevalence of poor sleep quality was 67.3% (95% CI: 64.0, 70.5%) among the Chinese elderly in nursing homes. Multiple binary logistic regression results showed that participants with the following characteristics had an increased risk of poor sleep quality after adjustments for other confounders: being 70–79 years old (AOR: 1.78, 95% CI: 1.08, 2.92) or 80 years old and above (AOR: 2.67, 95% CI: 1.68, 4.24); having one to two kinds of chronic diseases (AOR: 2.05, 95% CI: 1.39, 3.01) or three or more kinds of chronic diseases (AOR: 2.35, 95% CI: 1.39, 4.00); depression symptoms (AOR: 1.08, 95% CI: 1.04, 1.11), anxiety

Xiangya School of Public Health of Central South
University (csugwxy@126.com).

**Funding:** The author(s) received no specific
funding for this work.

**Competing interests:** The authors have declared
that no competing interests exist.

symptoms (AOR: 1.11, 95% CI: 1.05, 1.18), and social support(AOR: 0.97, 95% CI: 0.95,
0.99). Additive interactions were detected between age and anxiety symptoms (AOR: 8.34,
95% CI: 4.43, 15.69); between chronic disease and anxiety symptoms (AOR: 8.61, 95% CI:
4.28, 17.31); and between social support and anxiety symptoms (AOR: 6.43, 95% CI: 3.22,
12.86).

## Conclusions

The prevalence of poor sleep quality among the elderly in nursing homes is relatively high.
Besides, anxiety symptoms has additive interactions with age, chronic disease and social
support for poor sleep quality. These findings have significant implications for interventions
that aim to improve sleep quality among elderly residents in nursing homes.

## Introduction

Ageing is a severe problem in China as well as throughout the world. According to the data of
the Sixth National Population Census in 2010, the number of people in China aged 60 years
and above was approximately 177 million, accounting for 13.26% of the total population [1].
The United Nations (UN) Commission on Population Development predicted that by 2020
the elderly population aged 60 and above in China would reach 243.8 million, accounting for
17.1% of the total population [2]. Sleep problems have become the most common complaints
among the elderly. Approximately 74% of the elderly men and 79% of the elderly women
reported sleep complaints in Italy [3]. In addition, previous studies reported that the preva-
lence of sleep disorders was approximately 30–40% among older residents [4,5]. However, as a
complicated phenomenon, the sleep quality of individuals is difficult to define and assess
objectively. Using a standardized assessment tool for sleep quality, the Pittsburgh Sleep Quality
Index (PSQI), epidemiological studies indicated that the self-rated prevalence of poor sleep
quality was 62.4% among older adults in Thailand [6], 16.6% among Mexican Americans aged
75 and older [7], 41.5% and 33.8% among the elderly adults living in urban and rural areas of
China, respectively [8,9]. However, several studies demonstrated that most sleep problems
may be exacerbated by institutional settings [10,11]. For instance, a study conducted on 100
selected individuals of over 65 years of age found that institutionalized elderly people pre-
sented more worse overall sleep quality and higher levels of daytime somnolence than non-
institutionalized elderly people [10]. Fetveit and colleagues [11] found that the prevalence of
sleep disturbance was approximately 70% among nursing home residents. In summary, sleep
problems are very common among the elderly and pose a major challenge to public health.

Evidence is emerging that poor sleep quality is positively associated with physical health,
mental disorders and quality of life [12,13]. For example, a large cohort study found that poor
sleep quality was associated with higher odds of hypertension among a Chinese rural popula-
tion [13]. Another large cohort study indicated that short sleepers with poor sleep quality had
an increased risk of cardiovascular disease [14]. In addition, many studies [15–17] have indi-
cated that poor sleep quality is a powerful predictor of suicidal ideation and depressive symp-
toms among the elderly. However, the underlying mechanisms between poor sleep quality and
these harmful consequences are still unclear.

To alleviate the personal suffering and adverse effects introduced by poor sleep quality, it is
essential to understand its prevalence pattern and associated factors. Previous studies have

demonstrated many factors related to poor sleep quality among older adults, including but not limited to the following four domains: (1) socio-demographic factors, such as age [9], marital status [18] and education [19]; (2) lifestyle factors, such as physical activity [20] and caffeine intake [21]; (3) emotional factors, such as stress [22] and depression [20]; and (4) chronic conditions, such as arthritis [9] and pulmonary disease [23]. However, few studies have examined the prevalence and risk factors of poor sleep quality in a nursing home population. Moreover, no study has explored the interactions of risk factors for poor sleep quality among elderly residents in nursing homes. Thus, the aims of this study were as follows: (1) to find the prevalence of poor sleep quality among the elderly in nursing homes in China; (2) to explore the risk factors for poor sleep quality; and (3) to explore the interactions of risk factors for poor sleep quality.

## Materials and methods

### Study population

This was a cross-sectional study conducted in the nursing homes of Hunan Province in China from October to December 2018. This study was approved by the Ethical Committee of Xiangya School of Public Health in Central South University (No.XYGW-2018-49). Written informed consent was obtained from all participants of this study. In order to select a representative elderly sample living in the nursing homes of Hunan Province, a multistage sampling method was used. First, each city from northern Hunan, southern Hunan and central Hunan was selected (i.e., based on geographical region): Changsha city, Hengyang city and Yiyang city, respectively. Subsequently, one county and two districts were randomly chosen from each selected city. For instance, Changsha County and the districts, Kaifu and Yuelu, were selected from Changsha City; Hengyang County and the districts, Yanfeng and Shigu, were selected from Hengyang City; and Yuanjiang County and the districts, Ziyang and Heshan, were selected from Yiyang City. Second, two townships were randomly chosen from each county. For example, Xingsha and Tiaoma were sampled from Changsha County; Xidu and Jingtou were sampled from Hengyang County; and Qionghu and Caowei were sampled from Yuanjiang County. Finally, two nursing homes were randomly selected from each selected district and township, and a total of 24 nursing homes were ultimately selected.

Residents in the selected nursing homes were included in our study if they met the following inclusion criteria: (1) had age 60 years and above; (2) had duration of entrance into nursing homes of more than one year; and (3) had physical and mental ability to participate in interviews. However, residents in the same homes were excluded if they (1) refused to participate in this study; (2) had a severe hearing impairment or language barrier, and (3) had a history of severe cognitive deficit. In summary, among a total of 2,055 older adults residents in the 24 nursing homes, 511 adults were excluded because they were younger than 60 years old or had stayed in a nursing home less than one year. A total of 603 older adults had a severe hearing impairment, language barrier or severe cognitive deficit, and 112 older adult residents who did not agree to participate were also excluded from this study. Of the remaining 829 older adults, 12 were excluded due to incomplete data. Ultimately, a total of 817 elderly adults were included in the data analysis in this study.

### Data measurement

All data were obtained through face-to-face interviews by trained staff who spent 30 to 60 minutes with each respondent when conducting the interviews. Data on the following variables were collected: socio-demographics (age, gender, education, marital status, monthly individual income, number of descendants, length of stay in a nursing home and medical insurance

status), lifestyle behavioural factors (smoking, alcohol drinking and physical exercise), physical status (activities of daily living (ADLs) and number of chronic disease) and social psychological factors (social support, depression symptoms and anxiety symptoms). Furthermore, marital status was dichotomized into stable and unstable, where unstable marital status included divorce, widow/widower and never having been married. Besides, smoking was assessed by a single item and defined as an average of at least one cigarette per day in the last year, and alcohol drinking was also assessed by a single item and defined as drinking a glass of wine per day in the last week. In addition, regular physical exercise was defined as to engage in at least 30 minutes of physical exercise at least three times a week. Moreover, ADL status was assessed using the Chinese version of Lawton and Brody's ADL scale, which subsequently dichotomized participants into disabled and normal [24]. Disabled ADL status was determined when participants had a total score of greater than 14 points. Cronbach's α of this instrument was 0.926 in this study. Additionally, self-reported chronic diseases were measured by a multiple choices question, which referred to chronic non-communicable diseases, including hypertension, diabetes, coronary heart disease (CHD), chronic obstructive pulmonary disease (COPD), hyperlipidaemia, stroke and others. Further, social support was measured using the Social Support Rating Scale (SSRS), which was developed by Xiao and is widely used in China. Using this instrument, participants were divided into two groups based on the median of the global scores (low social support: score≤30 points; and high social support: score>30 points). As regards depression symptoms, they were measured using the Chinese version of the Geriatric Depression Scale-30 items (GDS-30) [25]. This instrument has been commonly used in China and has indicated high validity and reliability among the Chinese elderly population[26]. Thus, when used on the sample of this study, a Cronbach's α of 0.925 was obtained. The GDS-30 consists of 30 true/false questions, and the total score ranges from 0 to 30 points. In this study, participants had depression symptoms if they scored 11 or more points on this instrument. On the other hand, anxiety symptoms were measured using the Chinese version of the Generalized Anxiety Disorder-7 scale (GAD-7). This instrument is a seven-item self-report instrument, in which each item assesses one of the typical symptoms of GAD over the last two weeks, thus, a total score ranges between 0 and 21. Cronbach's α of this tool was 0.933 in this study. Participants who scored 10 or more were considered to have anxiety symptoms in this study [27].

## Assessment of sleep quality

Sleep quality among the subjects was assessed using the Chinese version of the Pittsburgh Sleep Quality Index (PSQI) [28], which has been proven to have good validity and reliability among older adults [29]. The PSQI is a self-rated questionnaire that assesses sleep quality in the past month. It contains nineteen items grouped into seven components: subjective sleep quality, sleep latency, sleep duration, sleep efficiency, sleep disturbances, use of sleeping medication, and daytime dysfunction. Each component score ranges from 0 to 3, and the total score ranges from 0 to 21 points. Participants with a global PSQI score greater than 5 were defined as having poor sleep quality in this study. Cronbach's α for this tool was 0.781 in this study.

## Statistical analysis

Categorical variables were presented as frequencies and percentages. Continuous variables were presented as the mean±standard deviation (SD). The distribution of continuous variables across subgroups of categorical variables was compared using Student's t-test or one-way analysis of variance (ANOVA). The relationship between categorical variables was analysed using the Pearson chi-square test or the Mantel-Haenszel chi-square test. Multiple binary logistic

regression analysis was used to examine risk factors associated with poor sleep quality. The adjusted odds ratios (AORs) and 95% confidence intervals (CIs) were also estimated after controlling for other confounders. Microsoft Excel, developed by Andersson and colleagues [30], was used to detect and calculate the additive interaction of risk factors for poor sleep quality. The relative excess risk due to interaction (RERI) and the attributable proportion due to interaction (AP) of equal to 0, or the synergy index (S) of equal to 1, suggested no additive interaction between two factors. A $P<0.05$ and $P<0.025$ indicated statistically significant results for groups and multiple comparisons, respectively. All statistical analyses were performed in SAS 9.3 (SAS Institute Inc., Cary, NC, USA).

## Results

### Characteristics of the study sample

In total, 817 elderly residents living in nursing homes were included in this study. Among them, more than half of participants aged 80 years and above. In addition, 54.0% of the subjects were female, and 46.0% were male. Also, approximately 40% of the participants had completed less than 6 years of education, and a similar proportion had stable marriages. Moreover, approximately one-third of the total sample reported monthly individual incomes of more than 3,000 RMB. Also, a small proportion of cigarette smoking (16.2%), alcohol drinking (11.8%), and subjects that had no chronic diseases (24.1%) were observed among the participants. Furthermore, approximately 36.0% of the subjects had depressive symptoms, while 16.0% had anxiety symptoms.

There were statistically significant differences between poor sleepers and good sleepers in age, education, marital status, monthly individual income, length of stay, number of chronic diseases, depression symptoms, anxiety symptoms, social support and ADL status (all $P<0.05$). Poor sleepers were seemingly those who had older age, lower education, unstable marriages, lower monthly individual incomes, longer lengths of stay, at least one type of chronic disease, depression symptoms, anxiety symptoms, lower social support and disabled ADLs. The results in detail are presented in Table 1.

### Prevalence of poor sleep quality

The mean global PSQI score was 8.5 points overall (95% CI: 8.2, 8.8), 8.3 points in males (95% CI: 7.8, 8.8) and 8.7 points in females (95% CI: 8.2, 9.1). There was no statistically significant difference in the PSQI score between males and females ($P = 0.293$). The total PSQI score increased with increased age, for example, from 7.4 points (95% CI: 6.6, 8.2) in subjects aged 60–69 to 8.9 points (95% CI: 8.4, 9.3) in subjects aged ≥80 ($P = 0.010$). Additionally, 550 individuals had a total PSQI score of greater than 5, and the overall prevalence of poor sleep quality was 67.3% (95% CI: 64.0%, 70.5%). Nevertheless, the prevalence of poor sleep quality was 66.8% (95% CI: 61.7%, 71.0%) among males and 67.8% (95% CI: 63.5%, 72.1%) among females. The prevalence also increased with increased age, that is, the prevalence increased from 54.1% (95% CI: 45.9%, 63.0%) in participants aged 60–69 years to 72.7% (95% CI: 68.7%, 77.1%) in participants aged ≥80 years ($P$ for trend $<0.001$). The results are shown in Table 2.

### Components of sleep quality

Most of the participants went to bed at 9 PM and woke up at approximately 6 AM. The average bedtime of the participants was 9.1±3.5 hours, and the average sleep duration was 6.8±3.9 hours. Most subjects could not fall asleep within 30 minutes. The average sleep efficiency of the participants was 80.0%. Approximately 85% of the subjects never used sleep medication.

**Table 1. Characteristics of the elderly residents in the nursing homes.**

| Characteristic | Total (n = 817) | Poor sleeper (n = 550) | Good sleeper (n = 267) | P-value[†] |
|---|---|---|---|---|
| Age (years) | | | | |
| 60~ | 135(16.5) | 73(13.3) | 62(23.2) | <0.001 |
| 70~ | 232(28.4) | 150(27.3) | 82(30.7) | |
| 80~ | 450(55.1) | 327(59.4) | 123(46.1) | |
| Gender | | | | |
| Male | 376(46.0) | 251(45.6) | 125(46.8) | 0.751 |
| Female | 441(54.0) | 299(54.4) | 142(53.2) | |
| Education (years) | | | | |
| 0–6 | 364(44.6) | 263(47.8) | 101(37.8) | 0.004 |
| 7~ | 203(24.8) | 138(25.1) | 65(24.4) | |
| 10~ | 250(30.6) | 149(27.1) | 101(37.8) | |
| Marital status | | | | |
| Stable | 302(37.0) | 186(33.8) | 116(43.4) | 0.007 |
| Unstable | 515(63.0) | 364(66.2) | 151(56.6) | |
| Medical insurance | | | | |
| Yes | 766(93.8) | 517(94.0) | 249(93.3) | 0.681 |
| No | 51(6.2) | 33(6.0) | 18(6.7) | |
| Monthly individual income | | | | |
| ≤3,000 RMB | 568(69.5) | 404(73.5) | 164(61.4) | <0.001 |
| >3,000 RMB | 249(30.5) | 146(26.5) | 103(38.6) | |
| Length of stay (years) | | | | |
| 1~ | 444(54.3) | 284(51.6) | 160(59.9) | 0.026 |
| 3~ | 373(45.7) | 266(48.4) | 107(40.1) | |
| Have descendants | | | | |
| Yes | 746(91.3) | 496(90.2) | 250(93.6) | 0.100 |
| No | 71(8.7) | 54(9.8) | 17(6.4) | |
| Smoking | | | | |
| Yes | 132(16.2) | 82(14.9) | 50(18.7) | 0.164 |
| No | 685(83.8) | 468(85.1) | 217(81.3) | |
| Alcohol drinking | | | | |
| Yes | 96(11.8) | 60(10.9) | 36(13.5) | 0.284 |
| No | 721(88.2) | 490(89.1) | 231(76.5) | |
| Physical exercise | | | | |
| Regular | 353(43.2) | 233(42.4) | 120(44.9) | 0.485 |
| Irregular | 464(56.8) | 317(57.6) | 147(55.1) | |
| Number of chronic diseases | | | | |
| 0 | 197(24.1) | 96(17.5) | 101(37.8) | <0.001 |
| 1~ | 450(55.1) | 318(57.8) | 132(49.5) | |
| 3~ | 170(20.8) | 136(24.7) | 34(12.7) | |
| ADL status | | | | |
| Normal | 268(32.8) | 152(27.6) | 116(43.4) | <0.001 |
| Disabled | 549(67.2) | 398(72.4) | 151(56.6) | |
| Depression symptoms | | | | |
| Yes | 294(36.0) | 251(45.6) | 43(16.1) | <0.001 |
| No | 523(64.0) | 299(54.4) | 224(83.9) | |
| Anxiety symptoms | | | | |
| Yes | 136(16.6) | 124(22.5) | 12(4.5) | <0.001 |

(*Continued*)

**Table 1.** (Continued)

| Characteristic | Total (n = 817) | Poor sleeper (n = 550) | Good sleeper (n = 267) | P-value[†] |
|---|---|---|---|---|
| No | 681(83.4) | 426(77.5) | 255(95.5) | |
| Social support | | | | |
| Low | 435(53.2) | 337(61.3) | 98(36.7) | <0.001 |
| High | 382(46.8) | 213(38.7) | 169(63.3) | |

Data are presented as n (%). ADLs, activities of daily living.

† P-value was determined by the Pearson chi-square test.

The mean scores for subjective sleep quality, sleep disturbance and daytime dysfunction were 1.3±0.9, 1.2±0.6 and 1.4 ±1.1, respectively.

No statistically significant differences were observed between males and females on all components of sleep quality. More older residents than younger people have used sleep medication. Moreover, older adults have more daytime dysfunction than younger individuals do. No significant differences were obtained in sleep latency, sleep efficiency or sleep disturbance between age groups. The results are shown in Table 3.

### Risk factors for poor sleep quality

Multiple binary logistic regression analysis results suggested that having age between 70 and 79 years (AOR: 1.78, 95% CI: 1.08, 2.92) or 80 years and above (AOR: 2.67, 95% CI: 1.68, 4.24) increased the risk of poor sleep quality when compared with age between 60 and 69 years after adjustments were made for other confounding factors. Moreover, participants with one to two kinds of chronic diseases (AOR: 2.05, 95% CI: 1.39, 3.01), three or more kinds of chronic diseases (AOR: 2.35, 95%: 1.39, 4.00), depression symptoms (AOR: 1.08, 95% CI: 1.04, 1.11), anxiety symptoms (AOR: 1.11, 95% CI: 1.05, 1.18) and social support (AOR: 0.97, 95% CI: 0.95, 0.99) were more likely to report poor sleep quality after adjustments were made for other variables. The results are shown in Table 4.

### Interactions of risk factors for poor sleep quality

The Andersson's Excel was used to detect interactions between risk factors for sleep quality and calculate their effect size on poor sleep quality. Thus, interactions were detected between the following variables: older age and anxiety symptoms, with a RERI of 2.83 (95% CI: 0.13,

**Table 2. The PSQI global score and the prevalence of poor sleep quality among the elderly in the nursing homes stratified by age and gender.**

| | N | PSQI score Mean (95% CI) | P-value | Poor sleep quality[†]Prevalence (95% CI) | P-value |
|---|---|---|---|---|---|
| Total | 817 | 8.5(8.2, 8.8) | | 67.3(64.0, 70.5) | |
| Gender | | | | | |
| Male | 376 | 8.3(7.8, 8.8) | 0.293 | 66.8(61.7, 71.0) | 0.751 |
| Female | 441 | 8.7(8.2, 9.1) | | 67.8(63.5, 72.1) | |
| Age | | | | | |
| 60~ | 135 | 7.4(6.6, 8.2) | 0.010* | 54.1(45.9, 63.0) | <0.001* |
| 70~ | 232 | 8.4(7.8, 9.1) | | 64.7(58.6, 70.5) | |
| 80~ | 450 | 8.9(8.4, 9.3) | | 72.7(68.7, 77.1) | |

†Poor sleep quality was determined by PSQI score >5.

*P-value for trend.

**Table 3. Gender- and age-specific scores of sleep quality components measured by the Chinese version of the Pittsburgh Sleep Quality Index.**

| Sleep quality components | | Total (n = 817) | Gender | | | Age group (years) | | | |
|---|---|---|---|---|---|---|---|---|---|
| | | | Male (n = 376) | Female (n = 441) | P-value | 60–69 (n = 135) | 70–79 (n = 232) | ≥80 (n = 450) | P-value |
| Subjective sleep quality[†], mean ±SD | | 1.3±0.9 | 1.3±0.9 | 1.3±0.9 | 0.674 | 1.2±0.9 | 1.3±0.9 | 1.3±0.9 | 0.482 |
| Sleep latency, n(%) | ≤15 min | 236(28.9) | 116(30.9) | 120(27.2) | 0.200 | 51(37.8) | 63(27.2) | 122(27.1) | 0.161 |
| | 16~min | 252(30.8) | 121(32.2) | 131(29.7) | | 34(25.2) | 75(32.3) | 143(31.8) | |
| | 31~min | 329(40.3) | 139(36.9) | 190(43.1) | | 50(37.0) | 94(40.5) | 185(41.1) | |
| Sleep duration, mean±SD | | 6.8±3.9 | 6.8±3.1 | 6.8±4.4 | 0.908 | 6.9±2.1 | 6.7±2.7 | 6.9±4.7 | 0.759 |
| Sleep duration, n(%) | <6 h | 262(32.1) | 121(32.2) | 141(32.0) | 0.253 | 38(28.1) | 80(34.5) | 144(32.0) | 0.008 |
| | 6–7 h | 278(34.0) | 118(31.4) | 160(36.3) | | 38(28.1) | 67(28.9) | 173(38.4) | |
| | >7 h | 277(33.9) | 137(36.4) | 140(31.7) | | 59(43.8) | 85(36.6) | 133(29.6) | |
| Habitual sleep efficiency[*], mean ±SD | | 0.8±0.4 | 0.8±0.3 | 0.8±0.4 | 0.593 | 0.8±0.2 | 0.7±0.2 | 0.8±0.5 | 0.558 |
| Habitual sleep efficiency, n(%) | <65% | 270(33.1) | 125(33.2) | 145(32.8) | 0.803 | 41(30.4) | 80(34.4) | 149(33.1) | 0.075 |
| | 65–74% | 103(12.6) | 48(12.8) | 55(12.5) | | 14(10.4) | 28(12.1) | 61(13.6) | |
| | 75–84% | 135(16.5) | 57(15.2) | 78(17.7) | | 15(11.1) | 34(14.7) | 86(19.1) | |
| | >85% | 309(37.8) | 146(38.8) | 163(37.0) | | 65(48.1) | 90(38.8) | 154 (34.2) | |
| Sleep disturbances[‡], mean±SD | | 1.2±0.6 | 1.2±0.6 | 1.2±0.6 | 0.914 | 1.2±0.6 | 1.3±0.6 | 1.3±0.5 | 0.148 |
| Any sleep disturbance, n(%) | Yes | 248(30.4) | 112(29.8) | 136(30.8) | 0.745 | 38(28.1) | 72(31.0) | 138(30.7) | 0.826 |
| | No | 569(69.6) | 264(60.2) | 305(69.2) | | 97(71.9) | 160(69.0) | 312(69.3) | |
| Use of sleep medication, n(%) | Yes | 115(14.1) | 44(11.7) | 71(16.1) | 0.072 | 9(6.7) | 23(9.9) | 83(18.4) | <0.001 |
| | No | 702(85.9) | 332(88.3) | 370(83.9) | | 126(93.3) | 209(90.1) | 367(81.6) | |
| Daytime dysfunction[**], mean ±SD | | 1.4±1.1 | 1.4±1.1 | 1.5±1.0 | 0.620 | 1.2 ±1.0 | 1.5±1.1 | 1.5±1.0 | 0.037 |

SD, standard deviation.

†Score range 0 to 3; higher scores indicate poor subjective sleep quality

*Derived from the formula: hours of sleep/(get-up time–usual bedtime)×100%; score range from 0 to 1; higher scores indicate higher sleep efficiency

‡Derived from PSQI items 5b-5j; score range from 0 to 3; higher scores indicate more sleep disturbances

**Derived from PSQI item 8 and 9; score range from 0 to 3; higher scores indicate more daytime dysfunction.

5.52), an AP of 0.34 (95% CI: 0.13, 0.55) and an S of 1.63 (95% CI: 1.11, 2.39); chronic disease and anxiety symptoms, with a RERI of 3.32 (95% CI: 0.31, 6.33), an AP of 0.39 (95% CI: 0.29, 0.48) and an S of 1.77 (95% CI: 1.54, 2.04); and social support and anxiety symptoms, with a RERI of 2.96 (95% CI: 1.02, 4.90), an AP of 0.46 (95% CI: 0.38, 0.54) and an S of 2.20 (95% CI: 1.65, 2.93). No statistically significant interactions were found between other factors after adjusting for confounders. The results are shown in Table 5.

## Discussion

This study examined the prevalence of poor sleep quality and its risk factors among the elderly in nursing homes in China. Additionally, interactions of risk factors for poor sleep quality were explored. It was found that poor sleep quality affected nearly two-thirds (67.3%) of the elderly residents in nursing homes of this sample, and those with older age, chronic diseases, depression symptoms, anxiety symptoms and lower social support were the most likely to be affected by poor sleep quality. Moreover, additive interactions were detected between age and anxiety symptoms; and between education and anxiety symptoms.

**Table 4. Risk factors for poor sleep quality among the elderly in nursing homes.**

| Variable | B | SE | Wald | AOR(95% CI)[†] | P-value |
|---|---|---|---|---|---|
| Age (years) | | | | | |
| 60~ | | | | 1.00 | |
| 70~ | 0.58 | 0.25 | 5.15 | 1.78(1.08, 2.92) | 0.023 |
| 80~ | 0.98 | 0.24 | 17.37 | 2.67(1.68, 4.24) | <0.001 |
| Number of chronic diseases | | | | | |
| 0~ | | | | 1.00 | |
| 1~ | 0.72 | 0.20 | 13.31 | 2.05(1.39, 3.01) | <0.001 |
| 3~ | 0.86 | 0.27 | 10.01 | 2.35(1.39, 4.00) | <0.001 |
| Depression symptoms | 0.07 | 0.02 | 17.29 | 1.08(1.04,1.11) | <0.001 |
| Anxiety symptoms | 0.11 | 0.03 | 14.02 | 1.11(1.05,1.18) | <0.001 |
| Social support | -0.03 | 0.01 | 5.32 | 0.97(0.95,0.99) | 0.021 |

SE, standard error; AOR, adjusted odds ratio.

†Adjusted for gender, education, marital status, smoking, alcohol drinking, physical exercise, descendants, medical insurance, monthly individual income, length of stay and ADL status, as well as the variables in the table.

This large population-based study used a standardized and valid PSQI tool to measure the characteristics of sleep quality among elderly adults in nursing homes in China. Based on this approach, we found that the prevalence of poor sleep quality was 67.3% among elderly residents in nursing homes in China. This finding is comparable to those of other previous studies conducted in elderly adults living in nursing homes in Turkey [31,32]. However, with a similar population, the same institutional setting and measurement tool, Stefan et al [33] found that 54.5% of individuals were poor sleepers among 894 elderly adults in Zagreb nursing homes. Tsai et al [34] found that 46.4% of participants reported poor sleep quality among 196 elderly

**Table 5. Interactions between risk factors for poor sleep quality among the elderly in nursing homes.**

| Factor 1 | Factor 2 | AOR[†] | RERI | AP | S |
|---|---|---|---|---|---|
| Age (years) | Anxiety symptoms | | 2.83(0.13, 5.52) | 0.34(0.13, 0.55) | 1.63(1.11, 2.39) |
| 60–69 | No | 1.00 | | | |
| ≥70 | No | 2.09(1.01, 4.32) | | | |
| 60–69 | Yes | 4.43(2.13, 9.20) | | | |
| ≥70 | Yes | 8.34(4.43, 15.69) | | | |
| Chronic disease | Anxiety symptoms | | 3.32(0.31, 6.33) | 0.39(0.29, 0.48) | 1.77(1.54, 2.04) |
| No | No | 1.00 | | | |
| Yes | No | 2.51(1.92, 3.29) | | | |
| No | Yes | 3.79(1.82, 7.86) | | | |
| Yes | Yes | 8.61(4.28, 17.31) | | | |
| Social support | Anxiety symptoms | | 2.96(1.02, 4.90) | 0.46(0.38, 0.54) | 2.20(1.65, 2.93) |
| High | No | 1.00 | | | |
| Low | No | 1.61(1.10, 2.35) | | | |
| High | Yes | 2.87(1.38, 5.95) | | | |
| Low | Yes | 6.43(3.22, 12.86) | | | |

AOR, adjusted odds ratio; RERI, the relative excess risk due to interaction; AP, the attributable proportion due to interaction; S, the synergy index.

†Adjusted for gender, education, marital status, smoking, alcohol drinking, physical exercise, descendants, medical insurance, monthly individual income, length of stay, depression symptoms and ADL status, as well as the variables in the table.

nursing home residents in Taiwan. A reason for the variance in these estimates may be explained by the different inclusion criteria for the study sample, as well as the differences in facilities and medical care among different institutions. As mentioned in the introduction, some sleep problems may be exacerbated by institutional setting. Our study also indicates that the prevalence of poor sleep quality in nursing homes was higher than in community homes. For instance, only 33.8% of community-dwelling elderly adults in Yanggu County [8] and 27.7% in Deqing County [35] in China reported poor sleep quality. On the one hand, the high self-rated prevalence may result from residents in nursing homes being older (55.1% of participants were aged 80 years and above in this study) and having more chronic conditions (78.5% of participants have at least one kind of chronic disease) than elderly adults in the community [36]. On the other hand, an uncomfortable environment and care routines in the institution may not promote sleep [37]. Among the seven PSQI components, lower sleep efficiency (62.2%) was the most common, followed by longer sleep latency (40.3%) and sleep disturbances (30.4%). These results were consistent with those of previous studies [31,32]. As shown in our study, elderly adults suffered from several negative emotions (e.g., depression and anxiety) that may lead to longer sleep latency and lower sleep efficiency [38]. The main cause of sleep disturbances in nursing home settings may be nocturia in elderly adults, as well as noise and an uncomfortable environment [39]. However, the use of sleep medication was not prevalent in this population, and only 14.1% of individuals reported that they had taken sleep medication in the previous month. These results indicated that sleep problems in residents may not attract enough attention by administrative and healthcare staff in nursing homes. Otherwise, people had no awareness and believed that they were obtaining healthcare services for sleep problems in such a setting.

Consistent with previous epidemiological studies in the community, our study found that the prevalence of poor sleep quality was higher in older participants [9,40]. However, the findings of studies on the association between sleep quality and gender were conflicting. Many studies indicate that females are more prone than males to experience poor sleep quality because a higher proportion of females have lower socioeconomic status and are more susceptible to anxiety and depression [40,41]. In contrast, some studies indicate that males were more likely to report sleep problems than females [42,43]. In our study, no significant difference in prevalence was observed between males and females, which is consistent with the findings of a study conducted among elderly attendees of a primary care centre in Malaysia [43] and the findings of another study conducted in nursing homes in Taiwan [34]. The differences in these findings may be related to the differences in the study sample and research location. Specific chronic conditions, such as hypertension [44], heart disease [45], arthritis [9] and COPD [8], were found to be associated with poor sleep quality. In our current study, one interesting finding is that the risk of poor sleep quality was increased with an increasing number of chronic diseases. This result is in accordance with those of other previous studies. For example, a cross-sectional study performed among 16,680 residents aged 65 years in eight low- and middle-income countries demonstrated that poor health and a high number of comorbidities were associated with more sleep complaints [46]. Also, another population-based cross-sectional study among 5,107 adults in Japan suggested that sleep quality was directly proportional to the number of comorbid conditions in a subject [47]. Thus, the mechanism underlying the association between multiple chronic conditions and poor sleep quality requires investigation to find ways for improving sleep quality among the elderly with at least one chronic disease.

Additionally, like in the previous studies [48–50], this study found that depression and anxiety symptoms independently increased the risk of poor sleep quality. Similarly, Cho HJ et al [51] found that a greater level of depressive symptoms had increased odds of sleep latency (≥1 hour) among 3,051 participants aged 67 and older in the United States of America (USA).

Moreover, another study conducted among 2,040 elderly Koreans suggested that poor subjective sleep quality, longer sleep latency and frequent use of sleeping medication were independently associated with depression [52]. Also, Potvin et al [53] suggested that daytime sleepiness and sleep disturbances were significantly associated with anxiety. In addition, Press et al [54] reported that depressive symptoms were associated with decreased sleep satisfaction, while anxiety symptoms were associated with difficulty in falling asleep, waking up during the night and morning weakness. Therefore, these findings suggest that reducing depressive and anxiety symptoms in the elderly may help improve sleep quality.

Further, in this study, investigations on additive interactions of risk factors for sleep quality found that anxiety symptoms had additive interactions with older age, and chronic disease. That is, participants who had anxiety symptoms and older age, or anxiety symptoms and chronic disease were more likely to have poor sleep quality than those with anxiety symptoms and aged between 60 and 69, or anxiety symptoms and without chronic disease, respectively. This is in line with the findings of some previous studies that anxiety symptoms were more common among older people [55,56], and chronic diseases and anxiety symptoms occurred contemporaneously among the elderly [57,58]. These results, therefore, may partly be the reason for the occurrence of the foregoing interactions, which provide new insights into the prevention and treatment of sleep problems among the elderly.

We assessed social support using the SSRS, which is a self-rated questionnaire developed by Xiao and widely used in epidemiological studies in China. This perceived social support scale evaluates social relationships with friends, neighbours, colleagues and family members as well as the availability of functional support. Many studies consider social support to be an important factor in the maintenance of health status. Considering social support and sleep quality, this study found that lower level of social support increased the risk of poor sleep quality among the elderly in the nursing homes. This result was consistent with the findings of previous studies [59–61]. Specifically, Kishimoto et al [60] found that individuals with weak social support from spouses or family members were at a higher risk of sleep disturbances than their counterparts with strong social support. In addition, lower level of emotional social support was associated with more difficulties in initiating and maintaining sleep among the 998 elderly African-Americans [61]. In this regard, evidence has shown that social support may influence sleep quality by enabling a feeling of belonging, and protection from social isolation and negative emotions (e.g., loneliness, depression and anxiety), thus decreasing the incident risk of sleep disorders [62–64]. Therefore, improving social support among the elderly is a necessary cause in the prevention of sleeping problems among the elderly. Similar to our study results, people with anxiety had an increased risk of poor sleep quality, especially those with lower social support. Additionally, a stronger social network may contribute to individuals having multiple sources of health information, thus increasing the probability of obtaining information that may help maintain and promote healthy sleep behaviour [65,66].

Quality of life is a comprehensive and multidimensional condition that refers to an individual's perceived physical and mental health under the influence of illness, injury and treatment over time. Obviously, sleep is an important influence factor for quality of life among Chinese elderly adults. It may provide a new perspective for us to improve quality of life among elderly in nursing homes due to sleep quality improvement.

To the best of our knowledge, this is the first study in China to explore the risk factors and their interactions for poor sleep quality among the elderly in nursing homes. Given the large population-based randomly selected sample, and the higher response rate, the findings of this study may provide valuable information for improving sleep quality among the aging populations in nursing homes. From a clinical point of view, concern about mental health in residents and intervention and treatment for anxiety symptoms may be helpful in the improvement of

sleep quality in nursing homes. Additionally, the amelioration of the living environment and the provision of more social support were also important.

However, there are several limitations pertaining to this study that need to be highlighted. First, this study cannot establish a causal relationship between poor sleep quality and its associated risk factors due to the cross-sectional study design. For example, some studies also declared that poor sleep quality was positively associated with depression, anxiety and chronic diseases [67,68]. Second, all data were measurements of self-rating, which may introduce recall bias in the findings. Third, the limited sensitivity and specificity of the Chinese version of PSQI may lead to misclassifications of poor sleepers. Finally, the findings may not be generalized to the elderly populations that are not living in nursing homes.

## Conclusions

The prevalence of poor sleep quality in nursing homes in China is relatively high. Its risk factors included older age, chronic diseases, depression symptoms, anxiety symptoms and lower social support. Besides, anxiety symptoms have additive interactions with age, chronic disease and social support for poor sleep quality. These findings have significant implications for interventions that aim to improve sleep quality among the elderly residents in nursing homes.

## Acknowledgments

We thank all the elderly residents in the nursing homes who participated in this study for their cooperation.

## Author Contributions

**Conceptualization:** Huilan Xu.

**Formal analysis:** Xidi Zhu, Zhao Hu, Yunhan Yu.

**Investigation:** Xidi Zhu, Yu Nie, Tingting Zhu, Yunhan Yu.

**Writing – original draft:** Xidi Zhu, Zhao Hu.

**Writing – review & editing:** Atipatsa Chiwanda Kaminga, Huilan Xu.

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
