## [Decision Letter · Decision Letter 0]

30 Dec 2019

PONE-D-19-31759

The prevalence of poor sleep quality and associated risk factors among Chinese elderly adults in nursing homes: a cross-sectional study

PLOS ONE

Dear Dr Xu,

Thank you for submitting your manuscript to PLOS ONE. After careful consideration, we feel that it has merit but does not fully meet PLOS ONE’s publication criteria as it currently stands. Therefore, we invite you to submit a revised version of the manuscript that addresses the points raised during the review process.

We would appreciate receiving your revised manuscript by Feb 13 2020 11:59PM. To enhance the reproducibility of your results, we recommend that if applicable you deposit your laboratory protocols in protocols.io, where a protocol can be assigned its own identifier (DOI) such that it can be cited independently in the future. For instructions see: http://journals.plos.org/plosone/s/submission-guidelines#loc-laboratory-protocols

We look forward to receiving your revised manuscript.

Kind regards,

Stephan Doering, M.D.

Academic Editor

PLOS ONE

Journal Requirements:

2. Please provide additional details regarding participant consent. In the ethics statement in the Methods and online submission information, please ensure that you have specified (1) whether consent was suitably informed and (2) what type you obtained (for instance, written or verbal). If your study included minors under age 18, state whether you obtained consent from parents or guardians. If the need for consent was waived by the ethics committee, please include this information.

3. Please include additional information regarding the survey or questionnaire used in the study and ensure that you have provided sufficient details that others could replicate the analyses. If you developed and/or translated a questionnaire as part of this study and it is not under a copyright more restrictive than CC-BY, please include a copy, in both the original language and English, as Supporting Information.

6. Your ethics statement must appear in the Methods section of your manuscript. If your ethics statement is written in any section besides the Methods, please move it to the Methods section and delete it from any other section. Please also ensure that your ethics statement is included in your manuscript, as the ethics section of your online submission will not be published alongside your manuscript.

Reviewers' comments:

Reviewer's Responses to Questions

**Comments to the Author**

1. Is the manuscript technically sound, and do the data support the conclusions?

Reviewer #1: Yes

Reviewer #2: Yes

2. Has the statistical analysis been performed appropriately and rigorously? 

Reviewer #1: Yes

Reviewer #2: Yes

3. Have the authors made all data underlying the findings in their manuscript fully available?

Reviewer #1: Yes

Reviewer #2: Yes

4. Is the manuscript presented in an intelligible fashion and written in standard English?

Reviewer #1: Yes

Reviewer #2: Yes

5. Review Comments to the Author

Reviewer #1: This study presents the original research that meets all applicable standard in ethics. Statistics, and other analyses are describe in sufficient detail. This article using standard English. Conclusions are presented and are supported by the data.

Reviewer #2: I agree that sleep problems have become the most common complaints among elderly

adults and I believe the study presented by the Authors from China are important. In addition to that the study sample is large and therefore the correlations shown beetwen the variables are strong.

In my opinion it is a well written study. I only wonder why the Authors have chosen the eldrely with 60 age and above not from 65? and how did you judge the cognitive problesm as it is mentioned in the exculsion criteria?

6. PLOS authors have the option to publish the peer review history of their article (what does this mean?). If published, this will include your full peer review and any attached files.

Reviewer #1: No

Reviewer #2: No

---

## [Author Response · Author response to Decision Letter 0]

14 Jan 2020

Response to [PONE-D-19-31759]

Thank you very much for your valuable comments and critiques, which have improved the clarity of our manuscript to a greater extent. Accordingly, we have thoroughly revised our manuscript based on the editors’ and reviewers’ comments and suggestions. The changes are highlighted in red for added words, or strikethrough for deleted words. Additionally, minor language, grammatical and stylistic errors have been corrected. Our specific point-by-point responses to the comments and queries have been addressed and itemized as following:

Journal Requirements:

Answer: Thank you so much for your helpful comment. We have checked and ensured that our manuscript meets PLOS ONE’s style requirements.

2. Please provide additional details regarding participant consent. In the ethics statement in the Methods and online submission information, please ensure that you have specified (1) whether consent was suitably informed and (2) what type you obtained (for instance, written or verbal). If your study included minors under age 18, state whether you obtained consent from parents or guardians. If the need for consent was waived by the ethics committee, please include this information.

Answer: Thank you so much for your valuable comment. We have included the ethics statement in the Methods section as follows: This study was approved by the Ethics Committee of Xiangya School of Public Health of Central South University (No.XYGW-2018-49). Written informed consent was obtained from all participants of this study.

3. Please include additional information regarding the survey or questionnaire used in the study and ensure that you have provided sufficient details that others could replicate the analyses. If you developed and/or translated a questionnaire as part of this study and it is not under a copyright more restrictive than CC-BY, please include a copy, in both the original language and English, as Supporting Information.

Answer: Thank you so much for your valuable comment. As suggested, we have included additional information regarding the survey or questionnaires used in the study. However, we did not develop nor translate a questionnaire as part of this study. Actually, all the questionnaires used in this study are Chinese versions, which have good reliability and validity as indicated in previous publish related studies.

Answer: Thank you so much for this important information. We have since checked on this link about information on unacceptable data access restrictions.

Answer: Thank you so much for your valuable comments and suggestions. We provided our data availability statement as follows: Data for this study contain potentially identifying or sensitive patient information. Therefore, it would be available upon reasonable request from the Ethics Committee of Xiangya School of Public Health of Central South University (csugwxy@126.com).

Answer: Thank you so much for this important information. Accordingly, we have linked an ORCID ID for the corresponding author.

6. Your ethics statement must appear in the Methods section of your manuscript. If your ethics statement is written in any section besides the Methods, please move it to the Methods section and delete it from any other section. Please also ensure that your ethics statement is included in your manuscript, as the ethics section of your online submission will not be published alongside your manuscript.

Answer: Thank you so much for your valuable comment and suggestion. We have included our ethics statement in the Methods section as follows: This study was approved by the Ethics Committee of Xiangya School of Public Health of Central South University (No.XYGW-2018-49). Written informed consent was obtained from all participants of this study.

Comments to the Author

1. Is the manuscript technically sound, and do the data support the conclusions?

Reviewer #1: Yes

Reviewer #2: Yes

Answer: Thank you so much for being satisfied with the technical level of our manuscript.

2. Has the statistical analysis been performed appropriately and rigorously? 

Reviewer #1: Yes

Reviewer #2: Yes

Answer: Thank you so much for your satisfaction with the statistical analysis in our manuscript.

3. Have the authors made all data underlying the findings in their manuscript fully available?

Reviewer #1: Yes

Reviewer #2: Yes

Answer: Thank you so much for verifying that we provided the data availability statement as regards our manuscript.

4. Is the manuscript presented in an intelligible fashion and written in standard English?

Reviewer #1: Yes

Reviewer #2: Yes

Answer: Thank you so much for your satisfaction with the way our manuscript has been presented.

5. Review Comments to the Author

Reviewer #1: This study presents the original research that meets all applicable standard in ethics. Statistics, and other analyses are describe in sufficient detail. This article using standard English. Conclusions are presented and are supported by the data.

Answer: Thank you so much for approving the importance and quality of our study.

Reviewer #2: I agree that sleep problems have become the most common complaints among elderly adults and I believe the study presented by the Authors from China are important. In addition to that the study sample is large and therefore the correlations shown beetwen the variables are strong.

In my opinion it is a well written study. I only wonder why the Authors have chosen the eldrely with 60 age and above not from 65? and how did you judge the cognitive problesm as it is mentioned in the exculsion criteria?

Answer: Thank you so much for approving the importance and quality of our study. Also, thank you so much for your constructive questions regarding the sample selection and identification of cognitive problems in our manuscript. In China, the elderly age standard starting point is 60 years, which was provided by <Law of the People's Republic of China on Protection of the Rights and Interests of the Elderly> and National Bureau of Statistics of China. In addition, we have excluded participants who had a history of severe cognitive deficit based on the disease archives information provided by caregivers or health service providers in the respective nursing homes.

---

## [Decision Letter · Decision Letter 1]

13 Feb 2020

PONE-D-19-31759R1

The prevalence of poor sleep quality and associated risk factors among Chinese elderly adults in nursing homes: a cross-sectional study

PLOS ONE

Dear Dr. Xu,

Thank you for submitting your manuscript to PLOS ONE. After careful consideration, we feel that it has merit but does not fully meet PLOS ONE’s publication criteria as it currently stands. Therefore, we invite you to submit a revised version of the manuscript that addresses the points raised during the review process.

We would appreciate receiving your revised manuscript by March 12, 2020. To enhance the reproducibility of your results, we recommend that if applicable you deposit your laboratory protocols in protocols.io, where a protocol can be assigned its own identifier (DOI) such that it can be cited independently in the future. For instructions see: http://journals.plos.org/plosone/s/submission-guidelines#loc-laboratory-protocols

We look forward to receiving your revised manuscript.

Kind regards,

Stephan Doering, M.D.

Academic Editor

PLOS ONE

Reviewers' comments:

Reviewer's Responses to Questions

**Comments to the Author**

1. If the authors have adequately addressed your comments raised in a previous round of review and you feel that this manuscript is now acceptable for publication, you may indicate that here to bypass the “Comments to the Author” section, enter your conflict of interest statement in the “Confidential to Editor” section, and submit your "Accept" recommendation.

Reviewer #1: All comments have been addressed

Reviewer #3: (No Response)

2. Is the manuscript technically sound, and do the data support the conclusions?

Reviewer #1: Yes

Reviewer #3: Yes

3. Has the statistical analysis been performed appropriately and rigorously? 

Reviewer #1: No

Reviewer #3: Yes

4. Have the authors made all data underlying the findings in their manuscript fully available?

Reviewer #1: Yes

Reviewer #3: No

5. Is the manuscript presented in an intelligible fashion and written in standard English?

Reviewer #1: Yes

Reviewer #3: Yes

6. Review Comments to the Author

Reviewer #1: this study is good enough but it may need some improvement data such as more specific characteristic demography, analysis data, and explanation more about quality of life among Chinese elderly adults.

Reviewer #3: This is a cross-sectional study examining in 817 elderly residents in nursing homes the prevalence of poor sleep quality and its risk factors

The study confirms a relatively high prevalence of poor sleep quality in nursing homes, previously reported. Anxiety symptoms had additive interactions with age, chronic disease and social support for

poor sleep quality.

On the whole, I have a quite positive opinion on this cross-sectional study. It has a quite large sample size and methods seem mostly accurate.

Major concerns:

1. I understand but not completely agree on the reduction of scores of anxiety, depression and social support to categorical variables, using a cut off.

I ask to integrate the current analyses using the actual scores. Not necessarily, this request means a deletion of categorical classifications, but continuous variables should also be consider din the analyses

2. Due to the number of comparisons performed, a significance level set al p<0.05 should be adequately corrected for multiple comparisons

3. Due to the potential effects of illumination levels in nursing homes on sleep-wake activity rhythms (e.g., Ancoli‐Israel et al. , 9: 373-379, 2000). I ask to collect (if possible) this information in the considered nursing homes and include this variable in their analyses

7. PLOS authors have the option to publish the peer review history of their article (what does this mean?). If published, this will include your full peer review and any attached files.

Reviewer #1: Yes: My Name is Elmeida Effendy, Psychiatrist (Consultant)

Reviewer #3: No

---

## [Author Response · Author response to Decision Letter 1]

16 Mar 2020

Response to [PONE-D-19-31759R1]

Thank you very much for your valuable comments and critiques, which have improved the clarity of our manuscript to a greater extent. Accordingly, we have thoroughly revised our manuscript based on the editors’ and reviewers’ comments and suggestions. The changes are highlighted in red for added words, or strikethrough for deleted words. Additionally, minor language, grammatical and stylistic errors have been corrected. Our specific point-by-point responses to the comments and queries have been addressed and itemized as follows:

Reviewer #1: this study is good enough but it may need some improvement data such as more specific characteristic demography, analysis data, and explanation more about quality of life among Chinese elderly adults.

Answer: Thank you so much for your valuable comment. As suggested we have included and described more specific socio-demographic characteristics of the sample in relation to the study objectives. We did not dwell much on discussing quality of life of the elderly in the nursing homes because our main focus was on sleep quality, a component of quality life, and its associated factors. Thus, we justified why it was important to study this phenomenon. However, we agree that an explanation about the quality of life among the Chinese elderly people would highlight some insights in understanding some issues that would affect the living conditions of this population. Therefore, we have added a paragraph in the section of introduction to describe some facts about the quality of life of this group of people. Nevertheless, this study did not examine the quality of life of this population in general, but just a component of it, namely sleep quality. We thank you so much for your observation, which we have taken as a consideration to investigate the quality of life of this population in our future research.

Reviewer #3: This is a cross-sectional study examining in 817 elderly residents in nursing homes the prevalence of poor sleep quality and its risk factors

The study confirms a relatively high prevalence of poor sleep quality in nursing homes, previously reported. Anxiety symptoms had additive interactions with age, chronic disease and social support for poor sleep quality.

On the whole, I have a quite positive opinion on this cross-sectional study. It has a quite large sample size and methods seem mostly accurate.

Major concerns:

1. I understand but not completely agree on the reduction of scores of anxiety, depression and social support to categorical variables, using a cut off. I ask to integrate the current analyses using the actual scores. Not necessarily, this request means a deletion of categorical classifications, but continuous variables should also be consider din the analyses

Answer: Thank you so much for your valuable comment. It should be noted that the tools that we used to measure anxiety symptoms, depression symptoms and social support have high validity and reliability, and they are widely used to investigate anxiety, depression and social support among the Chinese elderly people. Besides, the cut-off points that we used on the total scores of these tools were adopted from other published studies on similar populations. However, according to the reviewer’s suggestion, we tried to consider anxiety, depression and social support as continuous variables, but the results did not deviate much from those when anxiety, depression and social support were considered as categorical variables (see the Table 1 below). Essentially, the results in the Table 1 below mean pretty much the same thing as the results in our original analysis .

Table 1 Risk factors associated with poor sleep quality among the elderly in nursing homes

Variable B SE Wald AOR(95% CI)† P-value

Age (years) 

60~ 1.00 

70~ 0.58 0.25 5.15 1.78(1.08,2.92) 0.023

80~ 0.98 0.24 17.37 2.67(1.68,4.24) <0.001

Number of chronic diseases 

0~ 

1~ 0.72 0.20 13.31 2.05(1.39,3.01) <0.001

3~ 0.86 0.27 10.01 2.35(1.39,4.00) 0.002

Depression symptoms 0.07 0.02 17.29 1.08(1.04,1.11) <0.001

Anxiety symptoms 0.11 0.03 14.02 1.11(1.05,1.18) <0.001

Social support -0.03 0.02 5.32 0.97(0.95,0.99) 0.021

2. Due to the number of comparisons performed, a significance level set al p<0.05 should be adequately corrected for multiple comparisons

Answer: Thank you so much for your valuable comment. A p<0.025 was set to indicate significant results for multiple comparisons in this study.

3. Due to the potential effects of illumination levels in nursing homes on sleep-wake activity rhythms (e.g., Ancoli‐Israel et al. , 9: 373-379, 2000). I ask to collect (if possible) this information in the considered nursing homes and include this variable in their analyses

Answer: Thank you so much for your valuable comment. We have not collected this information in our investigation; therefore we will consider this information in our further study.

---

## [Decision Letter · Decision Letter 2]

23 Apr 2020

The prevalence of poor sleep quality and associated risk factors among Chinese elderly adults in nursing homes: a cross-sectional study

PONE-D-19-31759R2

Dear Dr. Xu,

We are pleased to inform you that your manuscript has been judged scientifically suitable for publication and will be formally accepted for publication once it complies with all outstanding technical requirements.

With kind regards,

Stephan Doering, M.D.

Academic Editor

PLOS ONE

Reviewers' comments:

Reviewer's Responses to Questions

**Comments to the Author**

1. If the authors have adequately addressed your comments raised in a previous round of review and you feel that this manuscript is now acceptable for publication, you may indicate that here to bypass the “Comments to the Author” section, enter your conflict of interest statement in the “Confidential to Editor” section, and submit your "Accept" recommendation.

Reviewer #1: All comments have been addressed

Reviewer #3: (No Response)

2. Is the manuscript technically sound, and do the data support the conclusions?

Reviewer #1: Yes

Reviewer #3: Yes

3. Has the statistical analysis been performed appropriately and rigorously? 

Reviewer #1: Yes

Reviewer #3: Yes

4. Have the authors made all data underlying the findings in their manuscript fully available?

Reviewer #1: Yes

Reviewer #3: No

5. Is the manuscript presented in an intelligible fashion and written in standard English?

Reviewer #1: Yes

Reviewer #3: Yes

6. Review Comments to the Author

Reviewer #1: the author has improved the data, there has been better writing in this article. This article will be more interest if you choose update references.

Reviewer #3: The authors adequately responded to the points that I raised. It seems now acceptable for publication

7. PLOS authors have the option to publish the peer review history of their article (what does this mean?). If published, this will include your full peer review and any attached files.

Reviewer #1: No

Reviewer #3: No

---

## [Editor Report · Acceptance letter]

4 May 2020

PONE-D-19-31759R2 

The prevalence of poor sleep quality and associated risk factors among Chinese elderly adults in nursing homes: a cross-sectional study 

Dear Dr. Xu:

I am pleased to inform you that your manuscript has been deemed suitable for publication in PLOS ONE. Congratulations! Your manuscript is now with our production department. 

With kind regards,

on behalf of

Professor Stephan Doering 

Academic Editor

PLOS ONE